# A model for maxilloturbinate morphogenesis in seals

**Jonathan E. Kings**[1]*, **Lars P. Folkow**[2], **Øyvind Hammer**[3], **Signe Kjelstrup**[4], **Matthew J. Mason**[5], **Fengzhu Xiong**[5,6], **Eirik G. Flekkøy**[1,4]

1 PoreLab, Department of Physics, University of Oslo, Oslo, Norway, 2 Department of Arctic and Marine Biology, University of Tromsø - the Arctic University of Norway, Tromsø, Norway, 3 Natural History Museum, University of Oslo, Oslo, Norway, 4 PoreLab, Department of Chemistry, Norwegian University of Science and Technology NTNU, Trondheim, Norway, 5 Department of Physiology, Development & Neuroscience, University of Cambridge, Cambridge, United Kingdom, 6 Gurdon Institute, University of Cambridge, Cambridge, United Kingdom

* jonathan.kings@fys.uio.no

**Data availability statement:** Raw harp seal tomograms are available from doi.org/10.5281/zenodo.10401062. Model implementation and simulation viewer are

## Abstract

The nasal cavities of mammals contain the maxilloturbinate bones, which are involved in reducing heat and water losses. The maxilloturbinates of Arctic seals develop into particularly elaborate labyrinthine patterns, which are well adapted to retain heat and moisture from exhaled gas. These structures develop prenatally and continue to grow postnatally. The developmental mechanism of labyrinthine patterning is unknown. Here we report a model of maxilloturbinate pattern formation in prenatal and juvenile seals based on a simple algorithmic description and three key parameters: target turbinate porosity, characteristic ossification time scale, and typical gestation time scale. Under a small set of geometrical and physical rules, our model reproduces key features of the patterns observed in the turbinate structure of three seal species. To validate our model, we measure complexity, hydraulic diameter, backbone fractal dimension, and Horton-Strahler statistics for a rigorous quantitative comparison with actual tomograms of grey and harp seal skull specimens. Our model closely replicates the structural development of seal turbinates in these respects. Labyrinthine maxilloturbinate development may depend on the ability for neighbouring bone branches to detect and avoid each other, potentially through the mechanosensing of shear stresses from amniotic fluid and air flow.

## Introduction

Maxilloturbinates in polar and subpolar phocid seal species (Fig 1) develop into highly intricate dendritic patterns [1–4], which play an important role in heat and water homeostasis [5–7]. The bone structure is connected dorsolaterally in the rostral part of the nasal cavity, singly-rooted for most of its length on the maxillary bone on either side, with branches more densely packed towards the middle and tapering off rostrally and caudally [2,4]. The labyrinthine pattern displayed by the turbinate system is characterized by a constant length scale, important for heat exchange, underscored by its fractal dimension $D_f = 2$ [2], with gradual changes in density along the direction of fluid flow (Fig 2), manifesting as an anisotropic pattern.

available from doi.org/10.6084/m9.figshare.24948990 or the images from simulations are available from the authors upon request.

**Funding:** This work was supported by the Research Council of Norway through its Centers of Excellence funding scheme (project number 262644 awarded to EGF). See https://www.forskningsradet.no/en/apply-for-funding/funding-from-the-researchcouncil/sff/. The Research Council of Norway did not play a role in study design, data collection or analysis, decision to publish, or preparation of the manuscript. There was no additional external funding received for this study.

**Competing interests:** The authors have declared that no competing interests exist.

Although pinniped turbinate development *per se* has not been studied extensively [2], we are assuming here that mechanisms uncovered from developmental studies across other mammal groups, as summarised below, are conserved in pinniped development.

In pigs (*Sus scrofa*), longitudinal maxilloturbinate growth involves endochondral ossification which extends in the rostral direction [8]. Transversally, the maxilloturbinate branches are shown to grow appositionally at branch tips in both fruit bats (*Rousettus leschenaultii*) [9] and pigs [8]. Growth at the tips of the branches is consistent with early branching followed by postnatal branch length growth, proposed for grey seals by [2] on the basis of a comparison of neonatal and adult tomograms.

Growth of branched turbinates is bilaterally asymmetrical beyond the initial main branches [1,10]. Differences in maxilloturbinate pattern between right and left sides of the nasal cavity are visible in tomogram sections of a given individual, while differences in maxilloturbinate measurements have been documented between adult individuals of the same species [2,3]. Such evidence strongly suggests that the maxilloturbinate growth pattern within each half of the nasal cavity is not rigidly pre-programmed.

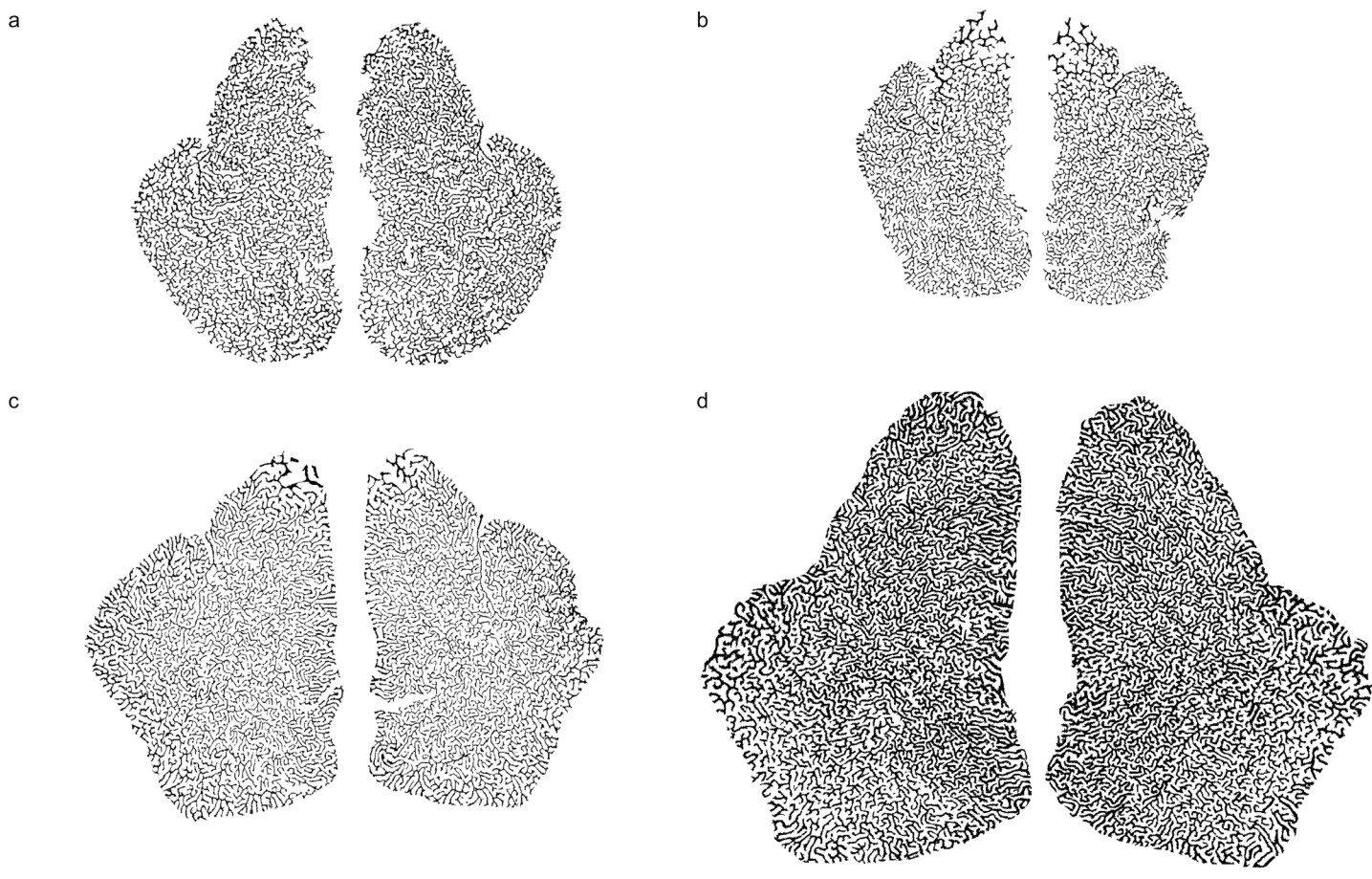

**Fig 1. Harp seal maxilloturbinate tomograms ordered from youngest to oldest.** The tomograms are reoriented through the densest part of the maxilloturbinate masses, and the specimens' relative ages are based on measured condylobasal lengths, CBL. Maxilloturbinate bones shown in black, airways in white. Enclosing nasal cavity walls not shown. $CBL^*$ values given below are relative to adult size ($CBL^* = 1$). Scale bar 50 mm. (a) Juvenile (specimen 7357), $CBL^* = 0.71$. (b) Juvenile (specimen 7498), $CBL^* = 0.78$. (c) Juvenile (specimen 7360), $CBL^* = 0.96$. (d) Adult (specimen 7495), $CBL^* = 1.0$.

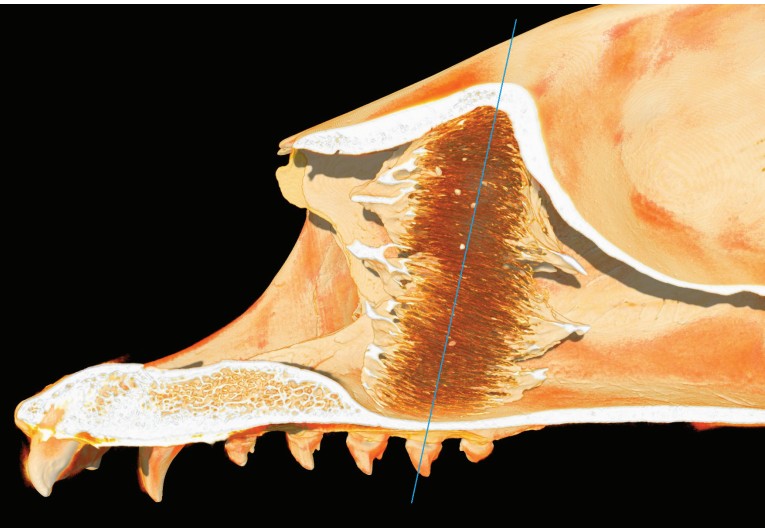

**Fig 2. CT reconstruction of the maxilloturbinate region of the adult harp seal (specimen 7495), sectioned parasagittally.** Blue line indicates approximate location of reoriented cross section reproduced in Fig 1d.

These key features of the pattern can shed light on its underlying developmental mechanisms. Morphological patterns in developing organisms often emerge as a consequence of physical and chemical interactions between the environment and cells, following a set of simple and limited local rules, making them productive subjects to study with mathematical models. In non-biological systems, model-based approaches have helped uncover an understanding of the underlying mechanisms of pattern formation, such as frictional fingering labyrinthine patterns in granular fluid dynamics [11,12], which bear a striking visual similarity to seal maxilloturbinate patterns. A prominent example in biology is the class of "Turing" models recapitulating the formation of periodic biological patterns via reaction-diffusion [13] and related activation-inhibition patterning mechanisms [14,15], such as in fingerprints [16] and zebra stripes [17]. We chose here an approach that involves a direct and continued role of mechanics in the development of turbinate structures, akin to pattern formation models that highlight the role of mechanical forces in directly shaping the tissue, such as in airway branching morphogenesis in mammalian lungs [18,19].

Following this approach, we introduce below a novel geometrical growth model to simulate seal maxilloturbinate pattern formation, with the goal of reproducing the most important structural features as simply as possible. Our approach begins with simple mechanistic assumptions that are subsequently tested by comparing the resulting simulated structures to tomographic images of turbinates at various developmental stages, using several quantitative measures. By identifying the essential requirements of our model, we are in a position to suggest biological mechanisms which may underlie seal maxilloturbinate development.

## Materials and methods

While transverse tomograms taken along the flow direction through the maxilloturbinate system show gradual variations from anterior to posterior, each consistently exhibits the characteristic labyrinthine patterns, preserving all key structural features. Therefore, we opted to model a representative two-dimensional cross-section in the transverse plane as an encapsulation of the full system, based on the following six assumptions:

1. The system grows within a space representing the nasal cavity, which becomes confining.
2. Growth occurs only at the tips of the structure, and branching may occur only via forking that happens preferentially in low density regions. We do not include any bone resorption in our model.
3. The structure has an elastic response that resists bending and locally minimizes its curvature.
4. As the structure grows, older parts become more rigid than new ones. The increase in rigidity occurs over a time scale comparable to the duration of the prenatal growth, with the rate decaying exponentially.
5. A self-avoidance mechanism keeps the branches from growing into each other.
6. All branches are established prenatally, with postnatal growth being limited solely to the elongation of pre-existing branches.

Our model describes a phenomenological implementation of these constraints by means of overdamped relaxation dynamics [20]. The interactions that drive this dynamical process are designed to reproduce the geometric effects of the underlying biological mechanisms involved but not necessarily to simulate these mechanisms directly. This is particularly true for the interaction that creates self-avoidance in the model.

In a two-dimensional transversal plane, the turbinate can be seen as a bony tree the branches of which display slight thickness variations. Abstracting away the thickness of the bone branches, we define the geometrical backbone of the maxilloturbinate as the set of thin lines in the centre of these branches, i.e. an infinitely thinned version of the bone's core. We thereby model the turbinate's backbone as a one-dimensional elastic tree of connected nodes embedded within a two-dimensional plane, with no inherent branch thickness.

The growing branches are constrained within a circular boundary of maximum radius $B$ corresponding to the nasal cavity (Fig 3). The node backbone tree is constructed iteratively by alternating two separate steps: a *relaxation* step and a *growth* step.

## Relaxation step

The relaxation step yields the position $\mathbf{r}_i$ of node $i \neq 0$ at integer time $t$ as

$$\mathbf{r}_i(t) = \mathbf{r}_i(t-1) + k(t_i)\,\mathbf{s}_i(t-1) \tag{1}$$

where $k(t_i)$ is a stiffness coefficient decaying to 0 over a characteristic rigidifying time scale $\tau$ with the time $t_i$ that has passed since the creation of the node $i$, as $k(t_i) = Ke^{-t_i/\tau}$, with $K$ constant. Note that stiffness is implemented by reducing $k$ thus reducing the displacement rate. Following assumption 4, $\tau = 4000$ is of the same order of magnitude as the total number of simulation iterations.

In order to achieve balance between the relevant forces, we define the displacement $\mathbf{s}_i$ to govern a spring force interaction acting on each node. This interaction is the sum of a repulsive displacement pushing nodes away from the boundary, a node-node force keeping nearest neighbours at a given separation, as well as repulsive non-neighbour interactions. The displacement introduced in Eq. (1) is defined as

$$\mathbf{s}_i = \mathbf{s}_i^b + \sum_{j\,\in\,\text{neighbours}(i)} \mathbf{s}_{ij}^n + \sum_{j\,\notin\,\text{neighbours}(i)} \mathbf{s}_{ij}^{nn}\,. \tag{2}$$

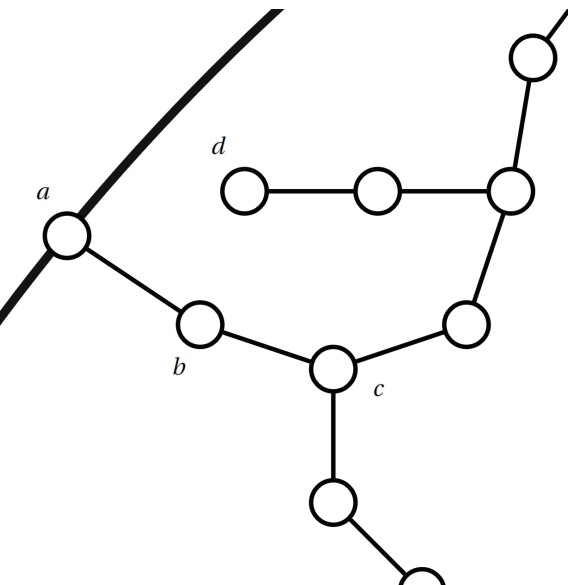

**Fig 3. We model the two-dimensional backbone of a maxilloturbinate mass as a series of connected nodes.** The nodes are constrained within a closed circular boundary representing the nasal cavity wall (thick line) of radius $B$. The first node ($a$) is fixed to the boundary. Neighbour nodes ($a$ and $b$, $b$ and $c$, etc.) are kept within a certain distance $r_0$ of each other by means of an overdamped spring force along the connection. Non-neighbour nodes ($b$ and $d$, $a$ and $c$, etc.) repel each other with a characteristic minimum distance $2r_0$. New nodes grow from branch tips (e.g. at $d$). Except for $a$, all nodes are repelled from the boundary.

These displacements along with Eq. (1) act to keep neighbour nodes such as $b$ and $c$ in Fig 3 within a characteristic distance $r_0$ of each other and to keep non-neighbour nodes such as $b$ and $d$ in Fig 3 away from each other:

$$\mathbf{s}_{ij}^n = \mathbf{e}_{ij}\left(\left|\Delta\mathbf{r}_{ij}\right| - r_0\right) \qquad \text{with } \Delta\mathbf{r}_{ij} \equiv \mathbf{r}_j - \mathbf{r}_i,\ \mathbf{e}_{ij} \equiv \frac{\Delta\mathbf{r}_{ij}}{\left|\Delta\mathbf{r}_{ij}\right|} \tag{3}$$

$$\mathbf{s}_{ij}^{nn} = \begin{cases} \mathbf{e}_{ij}\left(\left|\Delta\mathbf{r}_{ij}\right| - m_j\right) & \text{if } \left|\Delta\mathbf{r}_{ij}\right| < m_j \\ 0 & \text{otherwise,} \end{cases} \tag{4}$$

where $s_{ij}^{nn}$ acts on particle $i$. The non-neighbour interaction is purely repulsive and keeps non-neighbour nodes a distance of at least $m_j$ from each other, where we have defined $m_j = \max(2r_0, \left|\Delta\mathbf{r}_{jl}\right|)$. Here, $l$ is the furthest nearest neighbour on the chain of node $j$. This ensures self-avoidance between different pieces of the node tree, thus effectively closing gaps in the chain. The chosen value of $r_0$ only sets the numerical scale over which the pattern evolves, and as such does not influence the structure itself. Node $i = 0$ remains attached to the boundary throughout.

The boundary radius $b$ and the labyrinth of cross-sectional bone area $A_b$ define the porosity, i.e. the proportion of air within the outer circumference, $\phi = 1 - A_b/(\pi b^2)$. Soft tissue covering the maxilloturbinates can be considered to be included within the area $A_b$. Since the model only describes the backbone of the turbinate section and not the branch thickness, a constant thickness $w$ is chosen such that the overall porosity $\phi_{\text{target}}$ at the final timestep of the simulation, averaged across simulation runs, equals 0.75. In order to attain this target porosity

throughout the simulation, we replace $b$ by $b'$ where $\phi_{\text{target}} = 1 - A_b/(\pi b'^2)$. Using the above equation for $\phi$ to replace $A_b$ by $A_b = (1 - \phi)\pi b^2$, we may solve for $b'$ to get

$$b' = b\sqrt{\frac{1 - \phi}{1 - \phi_{\text{target}}}} \tag{5}$$

which is the new boundary radius, and increases up to a maximum of $B$. In our model the confining cavity therefore grows as a result of the developing turbinate system. Biologically, the causal relationship must be inverted, with turbinates developing within the enclosing cavity. However, this is simply an implementation detail: from a mathematical standpoint, identical results would be obtained if the sequence were reversed, with the enclosing boundary updated first and the turbinate as a result.

It should be noted that the chosen branch thickness $w$ is also used in the evaluation of metrics dependent on branch thickness and comparisons with experimental data below, as well as for visualization of the simulated tree. When visually comparing model results with tomograms, this adjustment should be borne in mind, along with the fact that the model does not attempt to account for age-related increases in bone thickness.

## Growth step

In order to model appositional branch tip growth, insertion of a new node is done either by extending a branch tip (e.g. from node $d$ in Fig 3) thus lengthening the branch, or by creating a fork at the tip of an existing branch. These node insertions throw the system off-equilibrium locally, requiring this growth step to be followed by several relaxation steps to restore a stable semi-equilibrium state before adding further nodes. The parent node, where growth takes place, is selected with a probability inversely proportional to local density (computed as proportional to the number of nodes within a radius $r_0$ of the candidate node). This gives priority to growth in low-density regions of the node tree, preventing high-density regions from becoming even denser. The position of the new node is chosen randomly on a circle of radius $r_0/10$ around the parent node. This small radius is chosen to prevent branch overlaps and to enable the new segment to grow to $r_0$ over time, with the balance of repulsive and attractive interactions leading to self-avoidance. Assumption 6 is integrated into the model by turning off forking after a certain threshold timestep $t_B$ corresponding to the time of the seal's birth, after which only growth by elongation of branch tips is possible.

## Comparison with experimental data and alternative branching models

We compared the structures resulting from this model with the geometry observed in tomograms obtained from grey seal (*Halichoerus grypus*) and harp seal (*Pagophilus groenlandicus*) skull specimens.

Grey seal specimens examined were a neonate (Cambridge Veterinary Anatomy Museum specimen SE1) and an adult (Oslo Natural History Museum specimen 7367), both scanned in the context of an earlier study [2]. Harp seal specimens examined were three juveniles of various sizes (specimens 7357, 7498, and 7360) and one adult (specimen 7495), all from the Oslo Natural History Museum and scanned in the context of this study, using a Nikon Metrology XT H 225 ST CT scanner with cubic voxel side length 33–59 μm. The processes used for scanning, denoising, and image analysis follow from [2].

The condylobasal lengths of the skull specimens are used as a proxy for developmental stage. We define the rescaled condylobasal length $CBL^* = CBL/CBL_{\text{adult}} \in [0, 1]$ to allow for

comparisons between the two species. The relative age of a seal $t^*$, where $t^* = 1$ corresponds to a fully developed adult, is defined as $t^* = CBL^*$ for ease of comparison with simulations. Note that in this definition, $t^* = 0$ does not correspond to a seal embryo in its first stage of development, but rather to a theoretical seal whose skull is infinitely reduced. The birth threshold timestep $t_B$ in our model is chosen as $t_B^* = CBL_B^*$, with $CBL_B^* \approx 0.68$ corresponding to the rescaled CBL of our juvenile grey seal specimen (SE1), known to be a neonate [2].

Model results are also compared to two alternative biological branching pattern formation models: Hannezo's branching and annihilating random walk model [21], and diffusion-limited aggregation (DLA; [22,23]). Simulation parameters for all three models were chosen such that the median final porosity obtained, $\phi_{\text{target}}$, equals 0.75. This value was chosen based on the porosity observed in tomograms, $\phi_{\text{obs}}$ (see section Observed tomogram porosity corresponds to the bony structure only, as porosity in a living specimen with soft tissue coating the maxilloturbinate bones must be lower. For simplicity, we assume here that the contribution from the mucosa is minimal.

All measurements below are shown as $x \pm \sigma$ in text, where $\sigma$ is the standard deviation across samples. Quantitative results for the three models are obtained as an average over 100 simulation runs.

## Results

The structures found in seal maxilloturbinates (Fig 1) are qualitatively similar to those displayed by our model (Fig 4), in terms of spatial distribution, branching patterns and arrangements, intricacy, regions of lower visual density, and meandering path progression. In particular, turning off forking after the gestation time scale $t_B^*$ leads to elongated branches at the edges of the system, similar to structures observed in adult seals (Fig 1d). Results from the two alternative models we consider (Fig 5) are characterized by low fractal dimensions $D_f < 2$ ($D_f = 1.71$ for the DLA model; [24]), with voids distributed over a wide range of length scales; our model produces a fixed channel width ($D_f = 2$), in agreement with turbinate observations.

In order to compare the structures quantitatively, we chose to examine porosity, hydraulic diameter and complexity, which have been measured to compare maxilloturbinates in seals and other mammals in previous work [2,25], as well as backbone fractal dimension and Strahler statistics, which provide fundamental properties commonly calculated in the geometrical analysis of dendritic and labyrinthine structures [12,26].

### Porosity measurements

Porosities found from the reoriented seal tomograms (without any mucosa attached) are $\phi = 0.71(4)$ ($n = 4$) in harp seals, and $\phi = 0.74(6)$ ($n = 2$) in grey seals. Throughout development, the data seem to indicate a slight increase in porosity in grey seals and a slight decrease in harp seals (Fig 6); these differences likely stem from intraspecific variations in porosity between individual specimens rather than being representative of trends across development in these species. The target porosity in our model was set to $\phi = 0.75$ as a direct result of these measurements and therefore remains relatively constant throughout simulation runs (Fig 6). The actual mean porosity throughout simulations is between 0.74 and 0.76. Root mean squared error (RMSE) between simulated and experimental porosities is $RMSE \approx 4.7 \times 10^{-2}$ for harp seals, $RMSE \approx 7.1 \times 10^{-2}$ for grey seals.

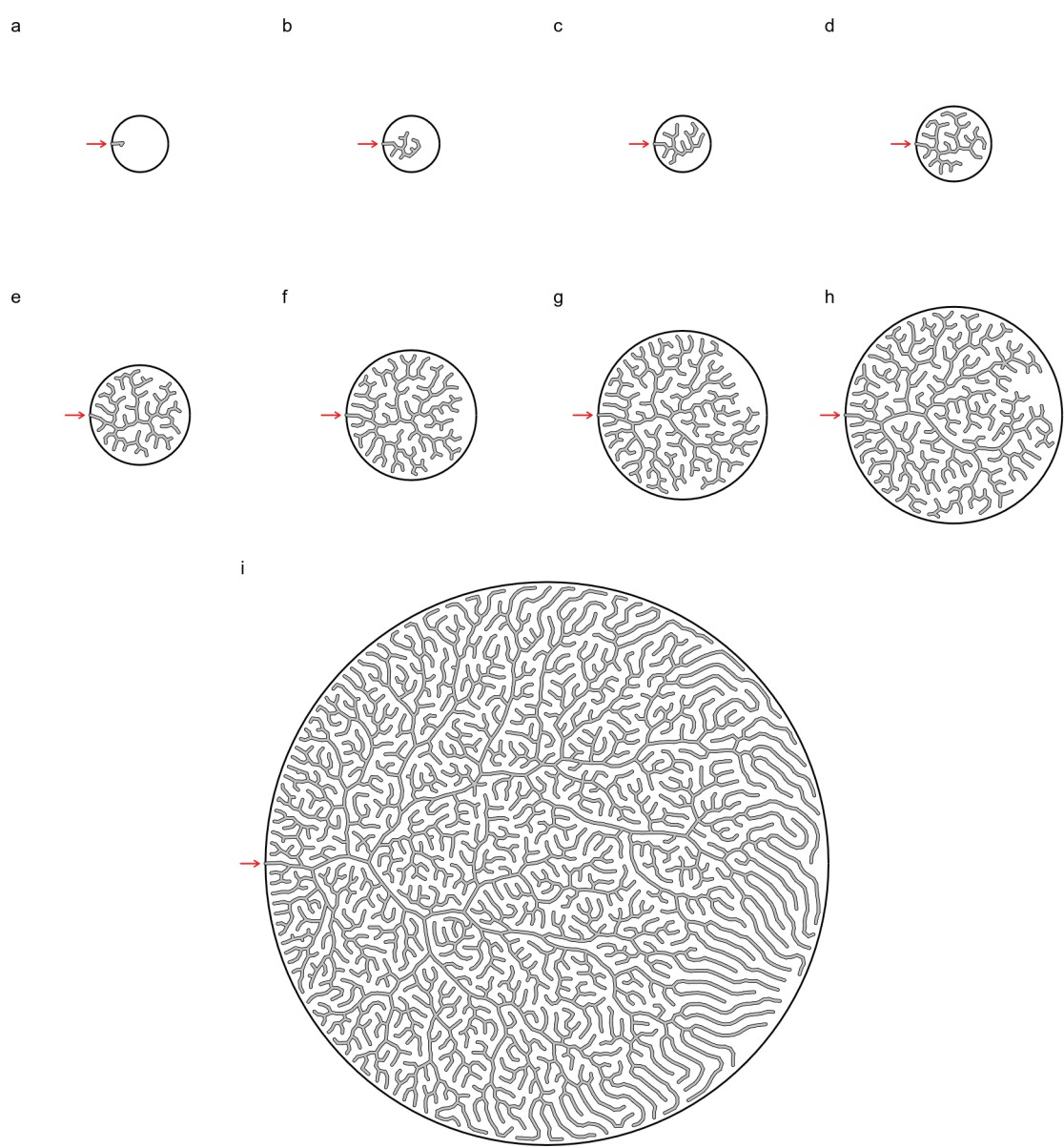

**Fig 4. Time series of modelled growth with our model.** Boundary circle (nasal cavity walls, with radius *B*) shown as a thick line. The node tree is rooted to the left side of the boundary (red arrows). The final result shows longer branches towards the right side and edges of the system due to forking being turned off and branches only elongating postnatally (i.e. beyond $t \geq t_B$). $t^*$ values correspond to rescaled simulation time (i.e. relative age, 0-1). (a) $t^* = 0.8 \times 10^{-2}$. (b) $t^* = 1.6 \times 10^{-2}$. (c) $t^* = 2.7 \times 10^{-2}$. (d) $t^* = 4.7 \times 10^{-2}$. (e) $t^* = 7.8 \times 10^{-2}$. (f) $t^* = 12.8 \times 10^{-2}$. (g) $t^* = 1.0$.

## Hydraulic diameter and complexity

The hydraulic diameter, $D_h$, represents the effective width of air channels, ignoring soft tissue. It is calculated as $D_h = 4A_c/P$, where $A_c$ is the total airway area through the chosen reoriented cross-section and $P$ the bone perimeter, and has dimensions of length. We define the rescaled hydraulic diameter as the dimensionless quantity $D_h^* = D_h/S$, where $S$ is the distance

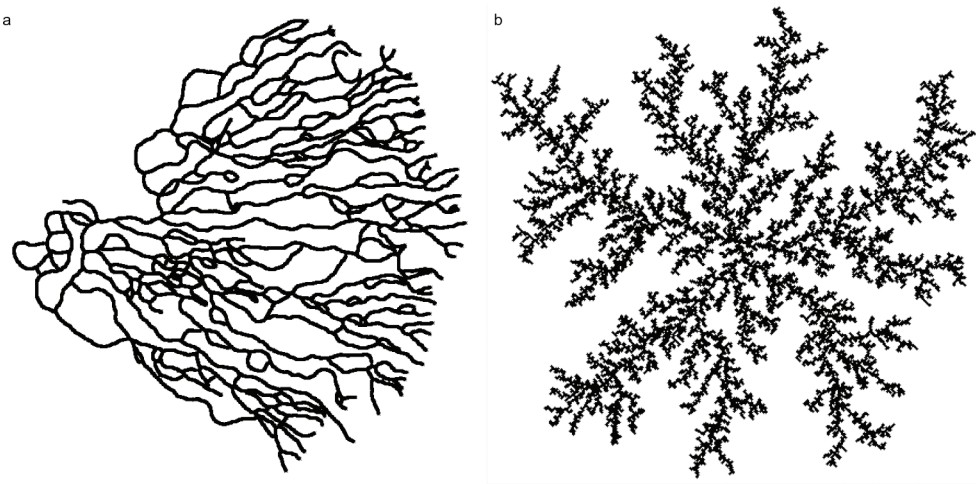

**Fig 5. Sample simulation results from alternative branch growth models.** (a) Hannezo's branching and annihilating random walk model [21]. (b) DLA [22].

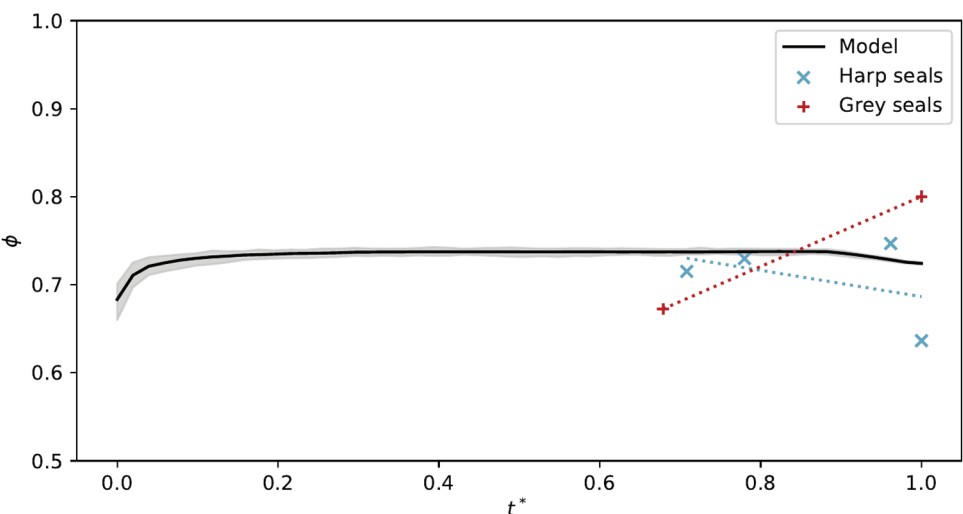

**Fig 6. Evolution of porosity ($\phi$) as a function of rescaled CBL (relative age; $t^*$) measured in seal skulls.** Dotted trend lines are fitted via linear regression of actual measurement values in harp seals (blue) and grey seals (red). The black line corresponds to the mean porosity computed through simulation runs, where $t^* = t/t_{\max}$ ($t$ in growth timesteps).

between the leftmost and rightmost points on the bone as seen in tomograms, to be compared with measurements on simulated bone labyrinths. In simulations, $S = 2B$. Hydraulic diameters in simulation outcomes from our model display a much lower standard deviation compared with experimental data (Table 1), but the modelled and experimental distributions are consistent in the following sense: $D_h^*$ tends to decrease as seals age in both examined species, which our model replicates to a reasonable degree under the $CBL^* = t^*$ assumption (Fig 7a) as evidenced by low root mean squared errors: $RMSE_{\mathrm{harp}} \approx 7.7 \times 10^{-3}$, $RMSE_{\mathrm{grey}} \approx 0.9 \times 10^{-3}$.

**Table 1. Measurements in seal tomograms and model results (mean ± standard deviation).**

| | | Harp seals | Grey seals | Our model | Hannezo model | DLA |
|---|---|---|---|---|---|---|
| Rescaled hydraulic diameter | $D_h^* \times 10^{-2}$ | 2.2±0.4 | 2.4±0.2 | 2.5 | 9±8 | 4.8±0.2 |
| Complexity | $C_x$ | 1.58±0.02 | 1.55±0.02 | 1.53 | 1.37±0.07 | 1.48±0.01 |
| Backbone fractal dimension | $d_m$ | 1.18±0.03 | | 1.12±0.03 | 1.08±0.06 | 1.10±0.02 |
| Strahler number | $\mathcal{H}$ | 6.6±0.5 | | 6.2±0.4 | 5.0±0.9 | 7.0±0.5 |
| Branching ratio | $R_B$ | 3.7±0.2 | | 3.6±0.3 | 4±1 | 3.8±0.4 |
| Branch length ratio | $R_L$ | 2.1±0.2 | | 1.8±0.2 | 2.2±0.8 | 2.0±0.3 |

The complexities ($C_x = \ln(P^2/4\pi)/\ln A_c$ treated as a dimensionless quantity) we measured in grey seal cross-sections are lower than those measured by [2] (Fig 7b). Measured complexities are greater in Arctic harp seals than in temperate grey seals, consistent with the findings of higher turbinate complexity in colder climate seal species [2]. Simulation tends to result in slightly lower complexity values (Fig 7b), which could be attributed to noise levels in scans artificially increasing perceived complexity. $RMSE_{\text{harp}} \approx 0.07$, $RMSE_{\text{grey}} \approx 0.04$.

The model's parameter space is limited: $r_0$ sets the numerical scale and $m$ the channel width. Variations in $\tau$ are found not to affect results significantly. Overall, the only free parameters that yield significant differences in results are the target porosity (in conjunction with the branch thickness $w$) and birth threshold timestep $t_B$, which can be chosen to match measurements from seal specimens.

## Backbone fractal dimension

The backbone dimension (or path dimension) $d_m$ of the backbone tree is defined as the slope obtained from the linear fit of the data points in a log-log plot, in which the bone to bone geodesic distances (walking along the branches) are plotted against the corresponding Euclidean distances [12].

Measuring the backbone dimension requires a representation of the maxilloturbinate cross-section as a connected graph, obtained from the image. While most of the scans are,

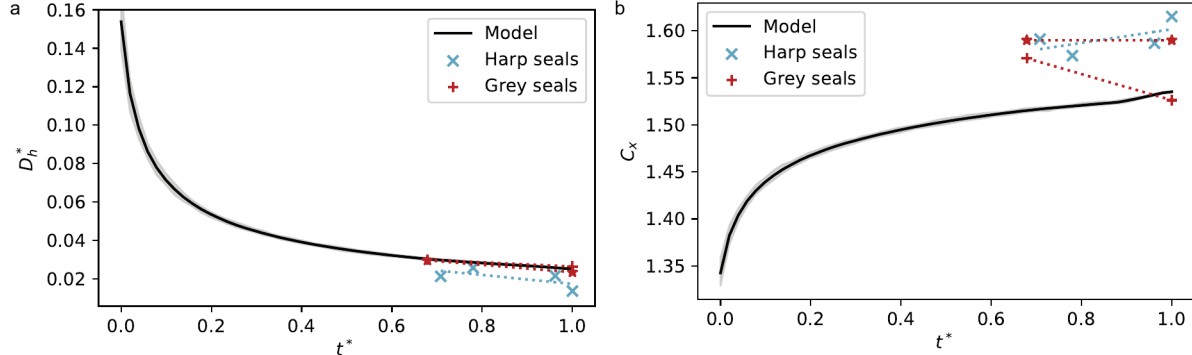

**Fig 7. Evolution of the rescaled hydraulic diameter ($D_h^*$) and complexity ($C_x$) in both simulations and seal tomograms as a function of simulation timestep and relative age ($t^*$).** Grey seal values (red stars) measured by [2] ($D_h$ values rescaled by a measured nasal cavity width of respectively 29.0 mm and 45.1 mm). (a) Rescaled hydraulic diameter. (b) Complexity.

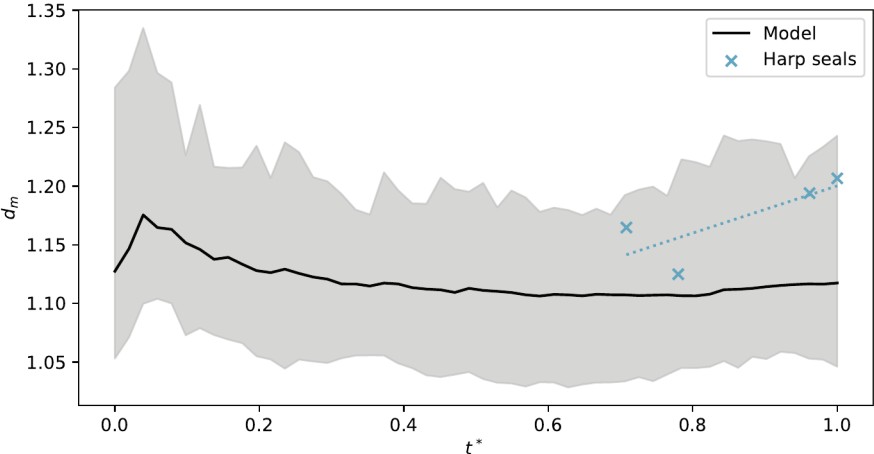

**Fig 8. Backbone dimension ($d_m$) as a function of relative age ($t^*$).** Shaded grey area represents the limits of $d_m$ computed at each timestep over 100 individual simulation runs.

after denoising, well-connected enough to avoid introducing significant errors (connectivity – ratio of area of the largest connected section of bone to the total turbinate bone area – between 92% and 99%), the adult grey seal tomograms are too noisy for the connectivity to reach any greater than 6%. Therefore, the adult grey seal specimen is excluded from this analysis (and similarly from the Strahler analysis, see below), and we only show results for the harp seal specimens.

The mean and standard deviation of the backbone fractal dimension are reasonably similar across harp seals and model results (Table 1). Relaxing our fourth assumption (that the structure rigidifies over time), by taking $\tau \to \infty$ in our model, results in early branches stretching as more new branches are added; those branches consequently have a lower backbone dimension.

While the backbone dimension appears to increase slightly as a function of age in seals while not noticeably doing so throughout simulations, the seal tomogram measurements fit reasonably well within bounds of measurements in model results (Fig 8), with a root mean squared error of $RMSE \approx 6.7 \times 10^{-2}$.

## Strahler analysis

Each of the individual branches in a turbinate cross-section can be assigned a Strahler order following a simple algorithm: leaf branches are assigned order 1 and pruned, leaving new leaf branches which can then be assigned order 2 and pruned, and so on until only the root branch is left [26–28]. Colour-coding branches by Strahler order yields Strahler diagrams for the various seal tomograms and model results, which display similarities such as the visual weight of the various colours. High-order (root) branches in seal scans tend to be more winding than in simulation outcomes in which the root branch can appear straighter (Fig 9). The Strahler number $\mathcal{H}$ [27] which denotes the order of the root branch, is consistently between 6 and 7 across both simulations and tomograms ($\mathcal{H}_{\text{model}} = 6.3(5)$, $\mathcal{H}_{\text{harp}} = 6.6(5)$), with similar mean, standard deviation, and scaling behaviour with respect to rescaled time ($RMSE \approx 0.56$, Fig 10a).

Also examined are the branching ratio $R_B$ and branch length ratio $R_L$, which are given by the slope of, respectively, number of branches and lengths of branches as a function of

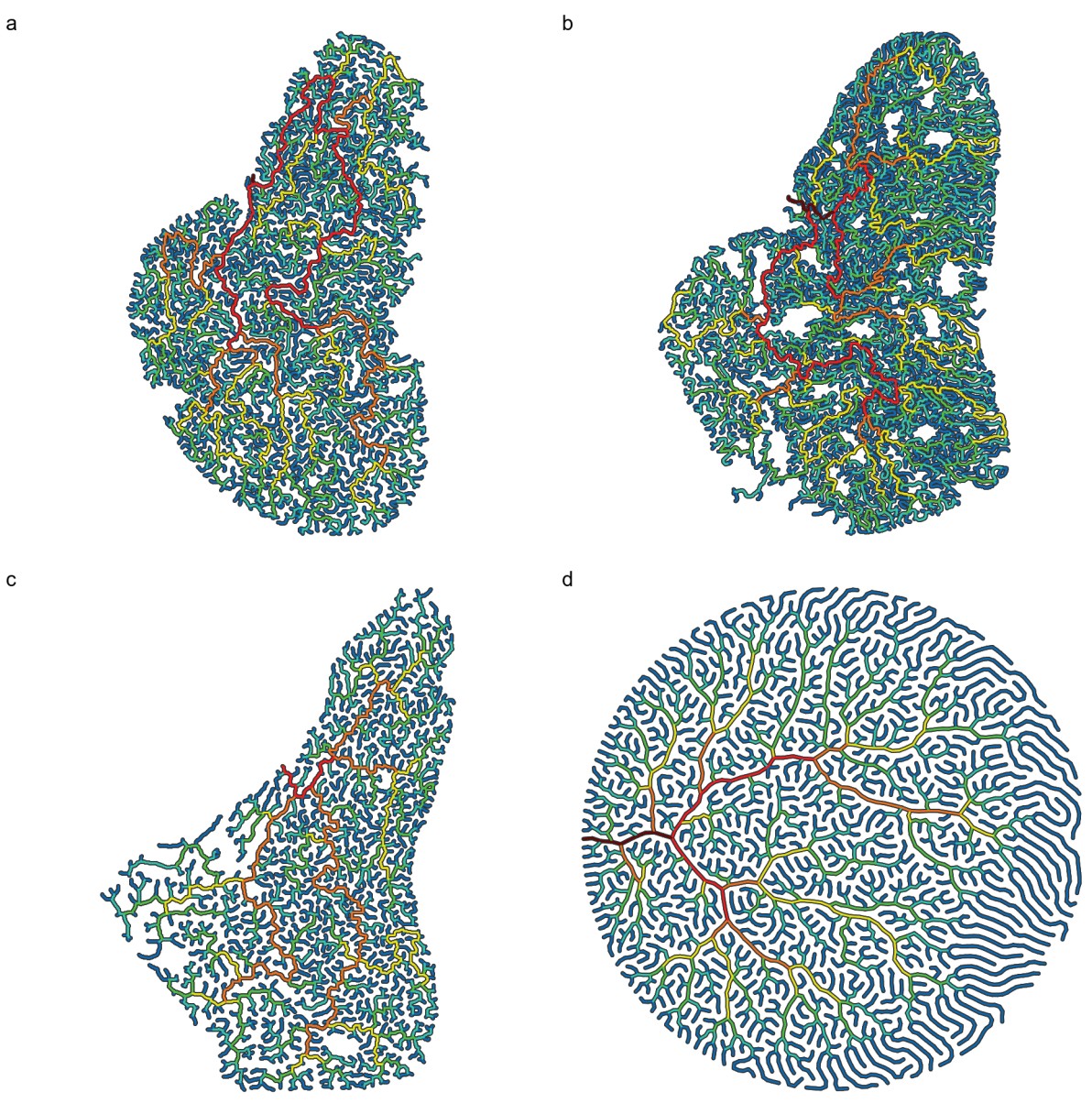

**Fig 9. Seal tomogram and simulation Strahler diagrams.** The root node (base of the root branch $w = \mathcal{H}$, in dark red) is selected in tomograms to correspond with the maxilloturbinate root. Branches with low Strahler orders are shown in colder colours ($w_{\text{blue}} = 1$), with warmer colours corresponding to higher orders. (a) Juvenile harp (specimen 7357). (b) Adult harp (specimen 7495). (c) Juvenile grey (specimen SE1). (d) Model

Strahler order [12,29–31]. These measurements reflect a geometrical aspect of the system that does not derive from any one parameter in our model – the only way to tweak the model in order significantly to affect the $R_B$ and $R_L$ measurements would be by encouraging growth in high-density areas. While $R_B$ is similar across simulations and tomograms, the length ratio distributions are more disparate (Table 1). Both remain roughly constant as a function of time/age in simulations and tomograms (Figs. 10b and 10c), with root mean squared errors of $RMSE \approx 0.26$ and $RMSE \approx 0.29$, respectively.

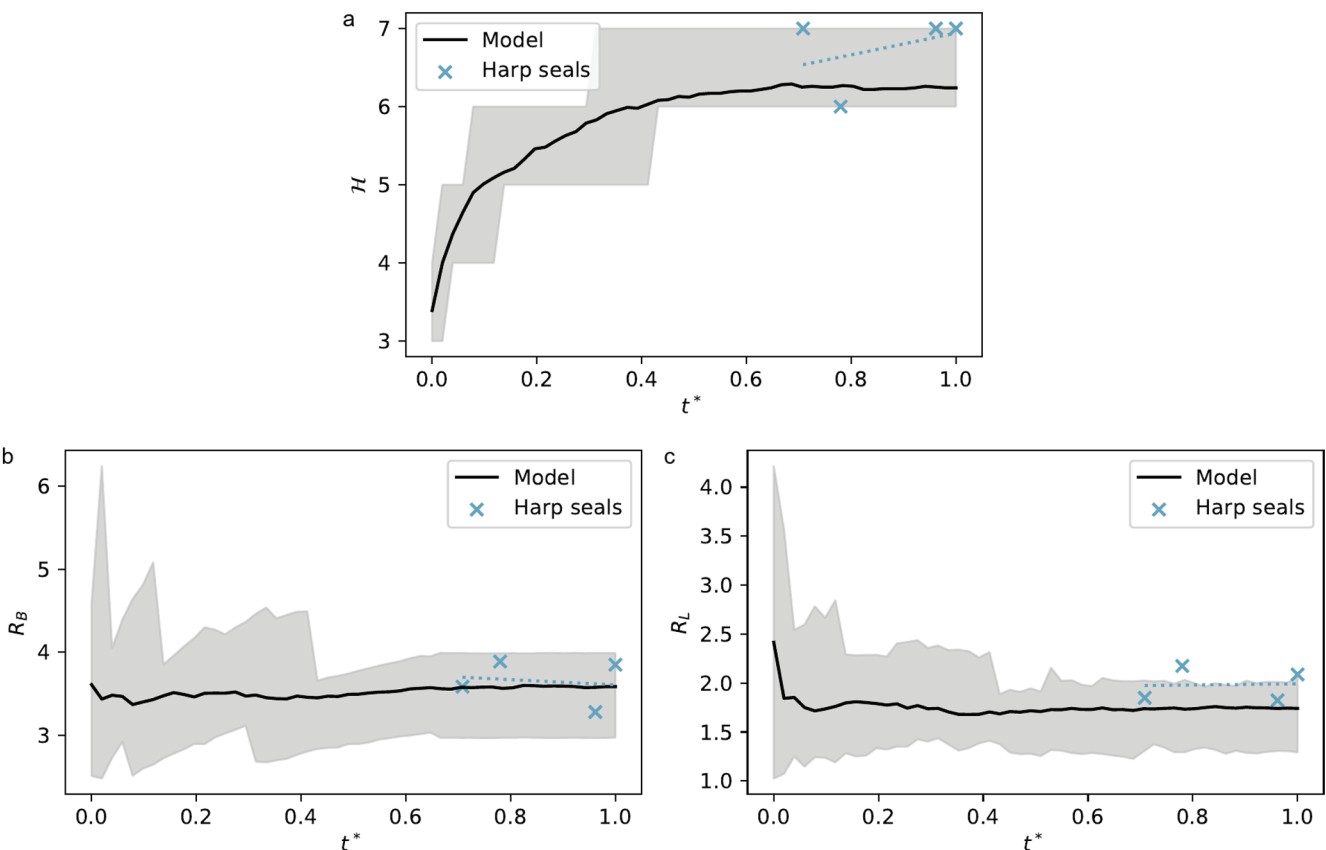

**Fig 10. Strahler number ($\mathcal{H}$), branching ratio ($R_B$) and length ratio ($R_L$) as functions of relative age ($t^*$).** Shaded grey area represents the limits computed at each timestep over 100 individual simulation runs. (a) Strahler number. (b) Branching ratio. (c) Branch length ratio.

## Model generalizability

Measurements made in specimens at different developmental stages and through simulation runs show that model timesteps and relative specimen ages (using the index of condylobasal length) are approximately linearly related (]Figs. 7a, 8, 10) with relatively low root mean squared errors. To test the generalizability of our model, we ran simulations with the greater porosity $\phi_{\mathrm{monk}} \approx 0.80$ observed in Mediterranean monk seal (*Monachus monachus*) maxilloturbinates as the target porosity (Fig 11). These yield results that closely match measurements made on monk seal tomograms (scans from [2]): a greater hydraulic diameter ($D^*_{h,\mathrm{model}} = 4.7 \times 10^{-2}$, $D^*_{h,\mathrm{monk}} = (4.6 \pm 0.2) \times 10^{-2}$) and lower complexity ($C_{x,\mathrm{model}} = 1.48 \pm 0.01$, $C_{x,\mathrm{monk}} = 1.52 \pm 0.01$) than in harp or grey seals, but similar backbone dimension ($d_{m,\mathrm{model}} = 1.12 \pm 0.04$, $d_{m,\mathrm{monk}} = 1.17 \pm 0.02$) and Strahler statistics ($\mathcal{H}_{\mathrm{model}} = 6.0 \pm 0.1$, $\mathcal{H}_{\mathrm{monk}} = 5.5 \pm 0.3$; $R_{B,\mathrm{model}} = 3.3 \pm 0.3$, $R_{B,\mathrm{monk}} = 3.8 \pm 0.3$; $R_{L,\mathrm{model}} = 1.7 \pm 0.2$, $R_{L,\mathrm{monk}} = 2.2 \pm 0.2$).

## Discussion

Overall, the rules (assumptions 1–6) embedded within our model yield results comparable with actual seal anatomy, and so may well correspond to biological mechanisms controlling turbinate morphogenesis within seals. This model thus allows us to formulate specific

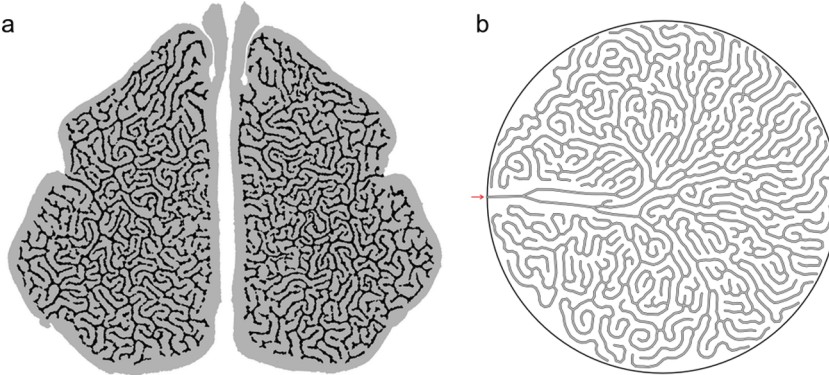

**Fig 11. Modelled development of a Mediterranean monk seal maxilloturbinate.** Monk seal (specimen NHMO M115) tomogram reproduced from [2] under the terms of the Creative Commons CC BY license (scale bar 50 mm), juxtaposed to final model result obtained for a target porosity $\phi_{monk}$ = 0.80. (a) Mediterranean monk seal. (b) Model.

hypotheses concerning how the intricate, labyrinthine pattern may develop, more precisely than other general branched pattern formation models.

Comparing the structure formed by our model (Fig 4) to the structures formed by the DLA model and the branching and annihilating random walk model of Hannezo et al. (Fig 5), we note a qualitative difference: while our model produces a characteristic, roughly constant channel width which is controlled by the interaction length $r_0$, the alternative models do not. The reason for this is that the structures of the DLA and Hannezo models do not include the deformations that in our case produce a well-defined porosity. Instead, they exhibit channel widths on a large range of length scales. The DLA model, in fact, produces fractals where the void space between branches will grow in an unlimited way with the structure size and, similarly, the Hannezo model lacks a compactification mechanism. For this reason they are inadequate to capture the relatively well-defined channel widths of the maxilloturbinate structures.

Assumption 1 of turbinate growth within a confined space is consistent with anatomical observations in other mammals, where the nasal cavity and the turbinate expand in size hand-in-hand, excepting early stages where the main branches grow within a relatively much larger chamber (e.g. [32]). In our model, this spatial confinement does not affect the key features of the pattern and the avoidance of branch growth into the boundaries can come from a similar biological mechanism as that between branches (assumption 5, discussed below).

Assumption 2, relating to tip growth and forking, was designed to be consistent with observations in other mammals [8,9]. Although bone resorption has been observed in the development of the scroll-like turbinate system of pigs, contributing to the curvature [8], seals have a dendritic turbinate pattern and resorption was not considered within our model. The progressive ossification of the maxilloturbinates from root to branch [9] will stiffen the branches over time, fixing the initial curvatures (assumptions 3 and 4). Branches nearer the root of the maxilloturbinate tree also tend to be thicker than terminal branches possibly as a result of growth, this being more obvious rostrally and caudally in the tomograms, supporting the notion of progressive stiffening. The separation of the branching stage and no-branching stage as prenatal and postnatal (assumption 6) was initially suggested by [2] based on comparison of grey seal tomograms and is supported here as juvenile harp seal samples show a similar branching depth and number as adults (Figs. 8 and 10b). One notable model implementation that controls the selection of the branching site is that it depends on density – a new branch will more

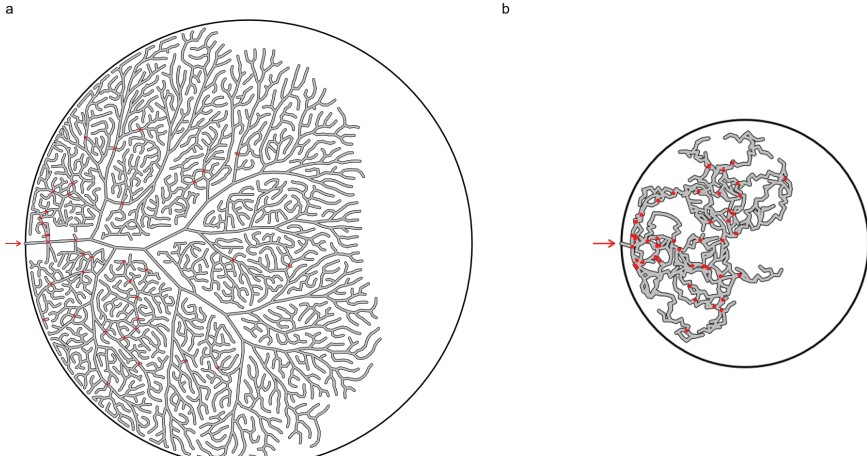

**Fig 12. Unstable model variants.** Simulation outcomes obtained for two variants of our model: the removal of density-based growth and the removal of branch avoidance. Red dots indicate points of collision between branches. Simulations were stopped beyond 50 branch intersections for clarity. (a) No density-based growth. (b) No branch avoidance.

likely appear in a region of low bone density than in a region already packed with branches. Without this constraint, our model yields uncharacteristic, uneven growth (Fig 12a). The ability of branching points to detect the density in neighbouring areas could be derived from a similar mechanism to the avoidance mechanism (assumption 5).

A key assumption of our model for which we have no direct evidence relates to branch avoidance (assumption 5). In the absence of an avoidance mechanism, appositionally growing lamellae would inevitably collide with each other (Fig 12b). Such a mechanism is consistent with the turbinate growth patterns observed in short-snouted dog (*Canis familiaris*) breeds, where insufficient space in the nasal cavity leads to turbinates growing into the pharynx [33]. The biological basis of this potential avoidance mechanism is currently unknown. How could a branch detect the presence of others across a length scale of hundreds of microns? Following the principles of Occam's razor, we would prefer to identify one feedback process suitable for guiding maxilloturbinate growth both prenatally and postnatally. There are unlikely to be any significant temperature gradients within the maxilloturbinate mass prenatally, given that the fetus is within the amniotic fluid at maternal core body temperature. A diffusing chemical signal seems unlikely postnatally, given the flow of gas through the turbinate system once the seal begins breathing air, which would sweep away any secreted morphogen. We therefore turn to considering mechanical signals for feedback purposes, which might operate both before and after birth. Turbinates are likely to be subject to shear stress in the longitudinal direction from amniotic fluid prenatally [34] or air postnatally. Blocking of postnatal airflow on one side can affect turbinate development in mice (*Mus musculus*) [35]. Coppola et al. estimated shear stress values in the mouse nasal cavity to exceed 1 Pa in some rostral positions, declining significantly towards the olfactory region posteriorly. They proposed that shear stress could be used as a mechanical signal to influence turbinate growth. We hypothesize that cells within the growing branches sense these shear stresses and respond by adjusting proliferation or growth rate. In a channel section of dimensions $\ell \times h \times b$ (Fig 13), the shear stress $\sigma$ on the walls in the direction opposite the pressure difference $\Delta P$ which is driving flow can be computed from the force balance $2\sigma S = A\Delta P$, with $A = b\ell$ the cross-sectional channel area and

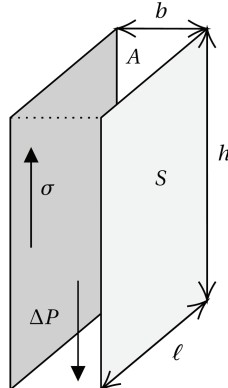

**Fig 13. Simplified geometry of a channel of dimensions $\ell \times h \times b$ with pressure gradient $\Delta P$ and shear stress on the walls $\sigma$.**

$S = h\ell$ the wall area. This yields $2\sigma = (b/h)\Delta P$; since $\Delta P$ is the same in neighbouring channels, it follows that $\sigma \propto b$, i.e. wider channels have a larger shear stress on the walls. In such larger channels within the maxilloturbinate mass (either between neighbouring branches or between a branch and the confining boundary), the greater shear stress may be sensed either by the bone directly (such as has been described by [36]) following deformation of the overlying mucosa, or indirectly via surface structures such as cilia (the role of which in mechanosensing has been discussed by [37]). Cilia are abundant on the pseudostratified columnar cells of the seal maxilloturbinate respiratory epithelium [4]. In the latter case, downstream cellular signalling involving different cell types would ultimately encourage local bone growth, resulting in space-filling by tip growth and new branch formation. Conversely, in narrow channels (i.e. higher-density areas), the shear stress becomes smaller with the flow and stops stimulating (or even starts inhibiting) cell growth, leading to the stabilization of the narrow channels and the prevention of branch collision and fusion. Additionally, the overall shape of the nasal cavity (narrower towards the anterior and posterior ends; Fig 2) may thereby encourage growth in the middle and inhibit growth at either end, resulting in the bone-dense central region and rostral and caudal tapering we observe.

Inspection of available CT scans of harp seals suggests that rostro-caudal growth of the maxilloturbinates occurs throughout development. We did not model growth of the maxilloturbinates in three dimensions to include the rostro-caudal dimension, and we can only speculate on how this extension might be regulated. Conrad et al. [38] found evidence to suggest that fluid flow within the prenatal mouse lung results in shear stress which could be driving anisotropic elongation of the epithelial tubes. It may be, then, that rostro-caudal expansion of the maxilloturbinates is similarly dependent on the direction of fluid flow, driven in this case by ventilatory movements. How directional cues might be extracted from this fluid flow and translated into rostro-caudal growth remains unknown. Elongation of the maxilloturbinates in the rostro-caudal direction would create longer channels, which could reduce the longitudinal pressure gradient $\Delta P/h$ and hence the shear stress experienced at any given point. Note that this proposed shear-stress mechanism is not suggested to model the growth at the anterior and posterior parts of the maxilloturbinate mass, but only the interior homogenization of channel widths.

Given such an influence of shear stress on bone growth, the overall porosity of the turbinates can become an optimized target for individual species to achieve through development, and a trait evolutionarily tunable by the genetic factors that regulate the gain in the mechanosensory system. This will produce the observed relatively stable porosity in a seal's nose throughout its postnatal development, which is variable between species but important for the heat and water exchange of the animal (e.g. [5,39]). Testing our hypothesis will require experimentation, for example altering the mechanical stresses experienced by the turbinates during prenatal growth, and then examining the genetic and cellular responses of the tissue. These experimental tests may be possible in a more accessible carnivore model, such as canine or feline, in which the turbinates likely share similar patterning mechanisms albeit having a less complex pattern [40]. Direct tests on the seals would best be approached with an in vitro culture model, where turbinate branch tip cells are grown and differentiated within controllable fluid environments.

## Acknowledgments

We thank Arnoldus Schytte Blix and Léa Wenger for helping to produce some of the CT scan data used in this study.

## Author contributions

**Conceptualization:** Jonathan Edward Kings, Lars P. Folkow, Signe Kjelstrup, Matthew J. Mason, Fengzhu Xiong, Eirik G. Flekkøy.

**Data curation:** Jonathan Edward Kings, Øyvind Hammer.

**Formal analysis:** Jonathan Edward Kings.

**Funding acquisition:** Eirik G. Flekkøy.

**Investigation:** Jonathan Edward Kings.

**Methodology:** Jonathan Edward Kings, Matthew J Mason, Fengzhu Xiong, Eirik G. Flekkøy.

**Project administration:** Eirik G. Flekkøy.

**Resources:** Øyvind Hammer, Matthew J. Mason.

**Software:** Jonathan Edward Kings.

**Supervision:** Eirik G. Flekkøy.

**Validation:** Matthew J. Mason, Fengzhu Xiong.

**Visualization:** Jonathan Edward Kings.

**Writing – original draft:** Jonathan Edward Kings.

**Writing – review & editing:** Lars P. Folkow, Øyvind Hammer, Signe Kjelstrup, Matthew J. Mason, Fengzhu Xiong, Eirik G. Flekkøy.

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
