## [Decision Letter · Decision Letter 0]

14 Aug 2024

PONE-D-24-24012 A model for maxilloturbinate morphogenesis in seals PLOS ONE

Dear Dr. Kings,

Thank you for submitting your manuscript to PLOS ONE. After careful consideration by two reviewers, we feel that it has merit but does not fully meet PLOS ONE’s publication criteria as it currently stands.   Therefore, we invite you to submit a revised version of the manuscript that addresses all points raised during the review process. Please also   - carefully explain your model assumptions as asked by reviewer #1 and especially how, in the actual 3D organ geometry, the anisotropy of plane sheet structures perpendicular to the considered 2D cross-section and branching exclusively within the considered 2D cross-section can be achieved.

- regarding the cited literature in the introduction, please see reviewer #1 and also relate to the early branching model by Hans Meinhardt (Models of Biological Pattern Formation, Academic Press, London, 1982, chapter 15). - make all data sharing links public at this point (reviewer #2).

We look forward to receiving your revised manuscript.

Kind regards,

Lutz Brusch, Ph.D.

Academic Editor

PLOS ONE

Journal Requirements:

3. Thank you for stating in your Funding Statement: "This work was partly supported by the Research Council of Norway through its Centers of Excellence funding scheme (project number 262644 awarded to EGF). See https://www.forskningsradet.no/en/apply-for-funding/funding-from-the-research-council/sff/. The Research Council of Norway did not play a role in study design, data collection or analysis, decision to publish, or preparation of the manuscript."

4. Thank you for stating the following in the Acknowledgments Section of your manuscript: "We thank Arnoldus Schytte Blix and L´ea Wenger for helping to produce some of the CT scan data used in this study. This work was partly supported by the Research Council of Norway through its Centers of Excellence funding scheme, project number 262644."

Please remove any funding-related text from the manuscript and let us know how you would like to update your Funding Statement. Currently, your Funding Statement reads as follows: "This work was partly supported by the Research Council of Norway through its Centers of Excellence funding scheme (project number 262644 awarded to EGF). See https://www.forskningsradet.no/en/apply-for-funding/funding-from-the-research-council/sff/. The Research Council of Norway did not play a role in study design, data collection or analysis, decision to publish, or preparation of the manuscript."

Reviewers' comments:

Reviewer's Responses to Questions

**Comments to the Author**

1. Is the manuscript technically sound, and do the data support the conclusions?

Reviewer #1: Partly

Reviewer #2: Yes

2. Has the statistical analysis been performed appropriately and rigorously? 

Reviewer #1: N/A

Reviewer #2: Yes

3. Have the authors made all data underlying the findings in their manuscript fully available?

Reviewer #1: Yes

Reviewer #2: No

4. Is the manuscript presented in an intelligible fashion and written in standard English?

Reviewer #1: Yes

Reviewer #2: Yes

5. Review Comments to the Author

Reviewer #1: Overall, I think this paper proposes an interesting simple model for maxilloturbinate morphogenesis that deserves to be published in principle. However, there are several shortcomings that I think would need to be addressed, as detailed below.

My main concerns are twofold:

1) Absence of direct biological evidence for the modeling assumptions. The rules by which the model evolves seems to be made to produce similar structures as observed in the tomography cross-sectional images, but these rules seem to be largely disconnected from any established mechanistic mechanism. The authors seem to be aware of this aspect, and the discussion at the end of the manuscript reflects this to some degree, but the present paper’s relevance to biologists will be indirect. However, there is value in simple models and their predictions, and I think that this should not preclude publication, if the uncertainties in the modeling assumptions are made clear and if speculation is clearly separated from evidence (this is to be improved, as detailed below).

2) Difficulty in translating the modeling assumptions, which may work in 2D, into a 3D growth mechanism. It is to be seen a weakness of the present manuscript that the model and analysis is limited to 2D, and that it remains unclear how the modeling assumptions are compatible with growth in 3D. A discussion of limitations of the model predictions arising from the dimensionality is missing. How do the authors picture the growth process in 3D? With the proposed model, one could imagine a quasi-2D growth pattern, extruded in the third direction orthogonal to the modeling plane, with the tree edge segments replaced by sheets of bone. (Is this a turbinate morphology that is consistent with 3D tomography images?) However, this would require a mechanism that tightly controls anisotropic growth and branching. Could the proposed mechanosensing of wall shear stress from amniotic fluid and air flow act as a directional cue, as it has been suggested for the biased growth of the lung epithelium? How would this be biologically implemented in the nasal cavity? I would expect at least a discussion of these aspects in the manuscript with clearly identified open questions, to avoid the present results from being overinterpreted.

Specific points to be addressed:

Given the uncertainty of the modeling assumptions (in particular #5 about self-avoidance), the last statement in the abstract (“… likely requires …”) seems overly asserting. “Likely” implies some form of probabilistic evidence, which is not given; instead, it is speculation. This should be weakened to something like “… in these respects, under the assumption that bone branches are able to detect and avoid each other through …”. For the same reason, the final introductory comment “biological mechanisms which are likely to underlie seal maxilloturbinate development” seems unjustified. This would be appropriate if “are likely to” was replaced by “may”, as it remains speculation.

The second half of the introduction, starting at line 21, is incoherent. Specifically:

Lines 21-25: The statements in this paragraph seem to be insufficiently linked with the cited literature. For instance, Reference 3 is cited to support the statement that “branched turbinate patterns are not identical between any two individuals of the same species”, but I am unable to find any evidence for this in the cited paper. Did I miss it? Reference 11 is cited to support the statement that “biological patterns tend to be at least partly algorithmically encoded rather than entirely pre-programmed”, but the cited paper is about a symmetry and simplicity bias in evolution, rather than stereotypicity or the genetic programming of entire structures.

It is not clear to me why the authors cite references 12 and 13 specifically out of the probably hundreds of papers (if not more) that study pattern formation in development mathematically, in lines 26-30. I don’t see how Zebra coat patterns (13) or a short Turing review based on a summer school lecture (12) would bear any particular relevance to the subject of the present manuscript.

Also the self-citations of references 14 and 15 appear picked out of numerous publications on labyrinthine patterns without apparent scientific reason. If the authors decide to maintain this selection, they should at least describe why these are more relevant to the present work than others.

Similarly, why did the authors cite reference 16 specifically? This seems like a random choice out of a large number of papers, and its relevance to the present work is not clear to me, particularly given that it is about magnetic fluids.

And finally, reference 21 is cited as an example of pattern formation models that study mechanical forces in epithelial invagination during branching morphogenesis in lungs. Reference 21 is a brief review, citing only two such works, and does not seem to be the appropriate source to cite, as it does not give credit to several more recent publications on the mechanics of branching morphogenesis.

Lines 53-61: What is the basis on which these assumptions are footed? Can the authors name and cite evidence in support of them? If there is none, these assumptions would need to be introduced more clearly as speculative and abstract (which would be fine, as long as this is made transparent).

Eqs. 1-4: One could argue that the vectors f are not forces (as named), but displacements of spring forces. The unit of f is that of a length, not of a force. Only after multiplication with the stiffness factor k, it becomes a force. This is only a terminological issue, but maybe the authors can think of a way to be more precise here.

How is A_b computed from the length of the backbone? There must be a turbinate diameter parameter that isn’t specified and that allows to go from length to area.

Lines 101-104: How is the local density computed? This information is missing from the manuscript.

Line 111: The rescaled quantities (with an asterisk in the superscript) are not yet defined at this point in the manuscript; their definition follows only later in the Results section (lines 149-156).

There are contradicting claims about the interpretation of uncertainty ranges in the manuscript. Line 134 says that all values are given as mean +- 2 sigma, but Table 1 says mean +- sigma.

Regarding the dismissal of the alternative models (Hannezo’s branching and annihilating random walk model and LDA), as shown in Fig. 4: Are these two alternatives free of tunable parameters that could potentially yield more realistic results if adjusted appropriately? If not, what effort did the authors make to adjust the parameters? The authors’ new model is a space-filling algorithm in 2D. From the steric interaction and the fact that branching is implemented to occur in low-density regions, it is clear that the fractal dimension of the resulting final labyrinth pattern will be 2. This is a trivial result, and similar growth rules combined with a co-expanding cavity could probably produce a space-filling structure also in the other two models.

Relatedly: In line 144, the authors claim that D_f<2 in the other two models, and D_f=2 in theirs. But what are the actual values of D_f? Did the authors quantify this? If yes, which method did the authors use? If not, how can the authors be certain that D_f<2 in the other two models?

Lines 177-179: Reference 2 does not seem to be an appropriate original source to cite for the definition of the hydraulic parameter. (Likewise for the complexity, line 188.) However, since the definition is given as well established, I don’t think any reference is needed for this.

Lines 177-178: How was the pore area A_c and the perimeter P measured, given that the simulated bone tree is infinitely thin?

Line 180: The definition of S is unclear to me. What do the authors mean by “leftmost” and “rightmost”? Relative to what? Within a pore, or globally such that S is almost 2B? If the latter is the case, does D*_h not simply measure the number of pores in the entire cavity?

The authors make the claim of “similar” or “consistent” distributions of measured quantities between simulations and tomography images (lines 184,208,227). I think these claims should be weakened, as there is no sufficient evidence for similarity of the entire distributions. The claims seem to be made based solely on the standard deviations.

How did the authors build a connected tree from the image shown in Fig. 10a to quantify the monk seal maxilloturbinates? Is this quantification reliable? This seems like a nontrivial, somewhat ambiguous task, given that the image is rather fragmented.

What is the quantitative effect of the rigidifying time scale τ on the model outcome? What value did the authors use? Is exponential bone stiffening a realistic model? A thorough discussion of this is missing.

How realistic is the model of a single anchoring point (root) of the bone tree? Does this reflect real maxilloturbinate morphogenesis, and would the model also work for multiple connections to the boundary?

Reviewer #2: This paper develops a model for the labyrinthine patterns of maxilloturbonate bones of different seal species. The model is a relatively simple 2D model that represents the pattern as a onedimensional collection of line segments. One growth rule governs the addition of new line segments and another rule governs the movement of line segments to incorporate growth, elastic behaviour and self-avoidance. The model results are qualitatively and quantitatively compared to real patterns using several quantifiers such as fractal dimension, porosity or Strahler number. Two additional simple models are additionally introduced and discarded as unrealistic.

The subject matter is interesting and the model is to my understanding novel. The assumptions of the model are biologically reasonable. The simplicity of the model with few parameters adds to its credibility. The paper is well-organized, generally easy to follow and the English is idiomatic and correct.

However, there are a few points of criticisms that should be addressed in a revision:

* The paper reduces the threedimensional bone structure to a twodimensional cross-section. This is clearly a reasonable strategy. However there are potential issues with this. For instance it could happen that the two-dimensional cross-section is not all connected anymore. Did this issue occur in the real specimen? Furthermore the choice of cross-section (more ventral or more dorsal) could influence the statistical properties.

* In equation (4) I find the notation r_{1,j} confusing, especially the index 1

* Equation (5) is used to update the new domain size. For this equation phi is used and this is calculated using A_b. It is mentioned that A_b is proportional to the backbone length but this is not sufficient to calculate A_b. I suspect that A_b = pi d^2 L where L is the backbone length and d is the radius of the backbone crosssection. This number d is clearly an important parameter as it controls the domain size and should be discussed in more detail.

* In the discussion of the grwoth stress it is mentioned that forking takes place inversely proportional to local density. How is the density defined? Clearly there must be some averaging procedure that is involved.

* The model uses only few parameters which is a strength of the model. It should nevertheless be discussed even if only briefly whether and to what degree the model results are robust to changes in the parameters.

* In the section ‘Data and code availability‘ the link toward the simulation viewer does not work.

* The image quality of the patterns in Figure 1 is not great.

Overall, I enjoyed reading the paper and I am confident that the raised points can be easily addressed.

6. PLOS authors have the option to publish the peer review history of their article (what does this mean?). If published, this will include your full peer review and any attached files.

Reviewer #1: No

Reviewer #2: **Yes: **Michael Kücken

---

## [Author Response · Author response to Decision Letter 1]

9 Oct 2024

Dear editor and reviewers,

Thank you very much for your thorough review. Edits have been made to the manuscript which should help answer the points you have raised. Please find below snippets copied from the original review for detailed answers to your individual comments. Please note that line numbers cited below correspond to the manuscript with tracked changes.

Reviewer #1: Absence of direct biological evidence for the modeling assumptions. The rules by which the model evolves seems to be made to produce similar structures as observed in the tomography cross-sectional images, but these rules seem to be largely disconnected from any established mechanistic mechanism. The authors seem to be aware of this aspect, and the discussion at the end of the manuscript reflects this to some degree, but the present paper’s relevance to biologists will be indirect. However, there is value in simple models and their predictions, and I think that this should not preclude publication, if the uncertainties in the modeling assumptions are made clear and if speculation is clearly separated from evidence (this is to be improved, as detailed below).

The logic surrounding our approach has been clarified with the addition of the sentences at the end of the introduction, lines 52-56. Edits have also been made throughout to more clearly separate speculation and assumptions from evidence.

Reviewer #1: Difficulty in translating the modeling assumptions, which may work in 2D, into a 3D growth mechanism. It is to be seen a weakness of the present manuscript that the model and analysis is limited to 2D, and that it remains unclear how the modeling assumptions are compatible with growth in 3D. A discussion of limitations of the model predictions arising from the dimensionality is missing. How do the authors picture the growth process in 3D? With the proposed model, one could imagine a quasi-2D growth pattern, extruded in the third direction orthogonal to the modeling plane, with the tree edge segments replaced by sheets of bone. (Is this a turbinate morphology that is consistent with 3D tomography images?) However, this would require a mechanism that tightly controls anisotropic growth and branching. Could the proposed mechanosensing of wall shear stress from amniotic fluid and air flow act as a directional cue, as it has been suggested for the biased growth of the lung epithelium? How would this be biologically implemented in the nasal cavity? I would expect at least a discussion of these aspects in the manuscript with clearly identified open questions, to avoid the present results from being overinterpreted.

The use of a 2D model assumes a geometry that is translationally invariant in the flow direction (see edit at lines 11-12), an assumption we have indeed made in the model. However, we note that the shear stress mechanism for self-avoidance in two dimensions also serves to explain the gradual changes in density along the flow direction seen in CT scans (Fig. 2): if a local bump in the side walls is encountered by the fluid flow, the streamlines will shift around it, causing a local shear stress drop compared to its surroundings within the channel, just as in the case where two channels of different widths are compared. Additionally, the conjectured effects of wall shear stress on growth are also sufficient to provide a directional cue as to whether the bone is anterior or posterior within the MT mass; see edits on lines 399-402 for clarifications.

Reviewer #1: Given the uncertainty of the modeling assumptions (in particular #5 about self-avoidance), the last statement in the abstract (“… likely requires …”) seems overly asserting. “Likely” implies some form of probabilistic evidence, which is not given; instead, it is speculation. This should be weakened to something like “… in these respects, under the assumption that bone branches are able to detect and avoid each other through …”. For the same reason, the final introductory comment “biological mechanisms which are likely to underlie seal maxilloturbinate development” seems unjustified. This would be appropriate if “are likely to” was replaced by “may”, as it remains speculation.

The statements cited have been amended; in particular see edits within the abstract and on line 57.

Reviewer #1: Lines 21-25 [24-30]: The statements in this paragraph seem to be insufficiently linked with the cited literature. For instance, Reference 3 is cited to support the statement that “branched turbinate patterns are not identical between any two individuals of the same species”, but I am unable to find any evidence for this in the cited paper. Did I miss it? Reference 11 is cited to support the statement that “biological patterns tend to be at least partly algorithmically encoded rather than entirely pre-programmed”, but the cited paper is about a symmetry and simplicity bias in evolution, rather than stereotypicity or the genetic programming of entire structures.

Claims made in this part of the introduction have been adjusted; see lines 24-32. Reference 2 (added line 28) shows illustrations of differences amongst grey seals and amongst mediterranean monk seals; reference 3 (kept) indicates that absolute and relative surface areas of olfactory and respiratory turbinates vary amongst individuals. Differences between the left and right side within a single individual have also been indicated, although different orientations of the sections make this difficult to prove.

Reviewer #1: It is not clear to me why the authors cite references 12 and 13 specifically out of the probably hundreds of papers (if not more) that study pattern formation in development mathematically, in lines 26-30 [33-37]. I don’t see how Zebra coat patterns (13) or a short Turing review based on a summer school lecture (12) would bear any particular relevance to the subject of the present manuscript.

These two references were indeed not particularly relevant and have been removed. The claim made lines 33-37 stands on its own without needing these examples.

Reviewer #1: Also the self-citations of references 14 and 15 appear picked out of numerous publications on labyrinthine patterns without apparent scientific reason. If the authors decide to maintain this selection, they should at least describe why these are more relevant to the present work than others.

References 14 and 15 have been kept, with an added explanation as to their relevance considering the resemblance of frictional fingering patterns to the present work (lines 37-41).

Reviewer #1: Similarly, why did the authors cite reference 16 specifically? This seems like a random choice out of a large number of papers, and its relevance to the present work is not clear to me, particularly given that it is about magnetic fluids.

The citation to reference 16 has been removed (line 40).

Reviewer #1: And finally, reference 21 [22] is cited as an example of pattern formation models that study mechanical forces in epithelial invagination during branching morphogenesis in lungs. Reference 21 is a brief review, citing only two such works, and does not seem to be the appropriate source to cite, as it does not give credit to several more recent publications on the mechanics of branching morphogenesis.

The citation to reference 21 (22 in the version with tracked changes) has been replaced with more appropriate references (23 & 24). The wording in the relevant section has been adjusted (see lines 41-49), and reference 18 (H. Meinhardt’s branching model) has been added.

Reviewer #1: Lines 53-61: What is the basis on which these assumptions are footed? Can the authors name and cite evidence in support of them? If there is none, these assumptions would need to be introduced more clearly as speculative and abstract (which would be fine, as long as this is made transparent).

Wording surrounding the 6 model assumptions has been adjusted to introduce them as speculative; lines 50-57 have been edited to clarify the rationale of the approach taken.

Reviewer #1: Eqs. 1-4: One could argue that the vectors f are not forces (as named), but displacements of spring forces. The unit of f is that of a length, not of a force. Only after multiplication with the stiffness factor k, it becomes a force. This is only a terminological issue, but maybe the authors can think of a way to be more precise here.

Wording concerning the f terms in these equations has been adjusted to better reflect what they represent; see lines 98-110 and 140.

Reviewer #1: How is A_b computed from the length of the backbone? There must be a turbinate diameter parameter that isn’t specified and that allows to go from length to area.

Reviewer #2: Equation (5) is used to update the new domain size. For this equation phi is used and this is calculated using A_b. It is mentioned that A_b is proportional to the backbone length but this is not sufficient to calculate A_b. I suspect that A_b = pi d^2 L where L is the backbone length and d is the radius of the backbone crosssection. This number d is clearly an important parameter as it controls the domain size and should be discussed in more detail.

A bone branch thickness would indeed be appropriate to compute the value of Ab, but is not relevant in the context of the simulations as the proportionality relation between Ab and the backbone length is sufficient. A branch thickness is needed however for calculations of Dh and Cx, which is now being discussed in more detail in lines 170-177.

Reviewer #1: Lines 101-104 [133-136]: How is the local density computed? This information is missing from the manuscript.

Reviewer #2: In the discussion of the grwoth stress it is mentioned that forking takes place inversely proportional to local density. How is the density defined? Clearly there must be some averaging procedure that is involved.

An explanation has been added lines 134-135.

Reviewer #1: Line 111 [144]: The rescaled quantities (with an asterisk in the superscript) are not yet defined at this point in the manuscript; their definition follows only later in the Results section (lines 149-156 [206-213]).

The definition has been moved from lines 206-213 to lines 160-166. The argument surrounding the chosen value for tB has been moved to the same section, lines 166-169.

Reviewer #1: There are contradicting claims about the interpretation of uncertainty ranges in the manuscript. Line 134 [191] says that all values are given as mean +- 2 sigma, but Table 1 says mean +- sigma.

Indeed; mean ± std.dev. is right; line 191 was a typo and has been corrected.

Reviewer #1: Regarding the dismissal of the alternative models (Hannezo’s branching and annihilating random walk model and LDA), as shown in Fig. 4 [5]: Are these two alternatives free of tunable parameters that could potentially yield more realistic results if adjusted appropriately? If not, what effort did the authors make to adjust the parameters? The authors’ new model is a space-filling algorithm in 2D. From the steric interaction and the fact that branching is implemented to occur in low-density regions, it is clear that the fractal dimension of the resulting final labyrinth pattern will be 2. This is a trivial result, and similar growth rules combined with a co-expanding cavity could probably produce a space-filling structure also in the other two models.

The DLA model has no free parameters. The original Hannezo model simulation code was used to arrive at the final parameters in the same manner as our own model: fine-tuning the available parameters until the best agreement was found with regards to porosity, and the results averaged over 100 runs of the simulation using those fine-tuned parameters. Df = 2 in the case of our model and the seal MTs is indeed trivial but has been deemed relevant in comparisons with other models for which this is not the case.

Reviewer #1: Relatedly: In line 144 [201], the authors claim that D_f<2 in the other two models, and D_f=2 in theirs. But what are the actual values of D_f? Did the authors quantify this? If yes, which method did the authors use? If not, how can the authors be certain that D_f<2 in the other two models?

The fractal dimension of the DLA model (Df = 1.71) has been added as an example, see line 201. In the case of the Hannezo model, Df can be quantified with the box-counting method, but varies greatly between 1 and 2 for different outputs of the model.

Reviewer #1: Lines 177-179 [234-236]: Reference 2 does not seem to be an appropriate original source to cite for the definition of the hydraulic parameter. (Likewise for the complexity, line 188 [246].) However, since the definition is given as well established, I don’t think any reference is needed for this.

These citations have been removed.

Reviewer #1: Lines 177-178 [234-235]: How was the pore area A_c and the perimeter P measured, given that the simulated bone tree is infinitely thin?

A constant branch thickness is chosen to match the porosity seen in seals and used for the measurements of our analysis that require a defined branch thickness. This has been clarified lines 170-177.

Reviewer #1: Line 180 [237]: The definition of S is unclear to me. What do the authors mean by “leftmost” and “rightmost”? Relative to what? Within a pore, or globally such that S is almost 2B? If the latter is the case, does D*_h not simply measure the number of pores in the entire cavity?

Lines 236-239 have been clarified. S=2B is indeed correct in the case of simulated labyrinths; the distinction is only relevant in the case of MT tomograms, for which there is no B as the nasal cavity is not circular.

Reviewer #1: The authors make the claim of “similar” or “consistent” distributions of measured quantities between simulations and tomography images (lines 184 [241], 208 [272], 227 [290]). I think these claims should be weakened, as there is no sufficient evidence for similarity of the entire distributions. The claims seem to be made based solely on the standard deviations.

These claims are made based on both the standard deviation and behaviour during growth (measured by the RMSE); the wording here has been clarified.

Reviewer #1: How did the authors build a connected tree from the image shown in Fig. 10a [11a] to quantify the monk seal maxilloturbinates? Is this quantification reliable? This seems like a nontrivial, somewhat ambiguous task, given that the image is rather fragmented.

Noise in the scans unfortunately does lead to rather fragmented trees when viewed as two dimensional images, but we know from the three dimensional structure that the trees formed by bone branches in cross sections are expected to be fully connected. Allowing for small gaps between connected sections in the scans up to a certain threshold distance yields almost fully-connected trees in all cases for the harp seals and monk seal; the adult grey seal scan however is still too fragmented in this regard, which is why it is excluded from discussions surrounding dm and Strahler analysis.

Reviewer #1: What is the quantitative effect of the rigidifying time scale τ on the model outcome? What value did the authors use? Is exponential bone stiffening a realistic model? A thorough discussion of this is missing.

The results are shown for a value of τ equal to a fifth of the total simulation iterations, but variations do not produce significant variation in results; this has been clarified line 254. That bone stiffens at an exponentially decaying rate in our model is an assumption (integrated into assumption 4, line 73).

Reviewer #1: How realistic is the model of a single anchoring point (root) of the bone tree? Does this reflect real maxilloturbinate morphogenesis, and would the model also work for multiple connections to the boundary?

Modelling the bone tree as rooted in a single anchoring point is consistent with observations; the MT is attached to the nasal cavity at a single point on the inner surface across most of its length (lines 4-9) which can be seen in some of the cross-sections (at the start of the thickest branch on the left in Fig. 1b, and on both sides in Fig. 11b, for example). Simulating two or more trees interacting within a single nasal cavity would produce similar results to the single-tree model

---

## [Decision Letter · Decision Letter 1]

29 Oct 2024

PONE-D-24-24012R1 A model for maxilloturbinate morphogenesis in seals PLOS ONE

Dear Dr. Kings,

Thank you for submitting your revised manuscript to PLOS ONE. After careful consideration by the same two reviewers, we conclude that still details are missing in the description of your model and some inconsistencies remain compared to your source code that currently limit the potential reproducibility of your otherwise very interesting study. Therefore, we invite you to submit a newly revised version of the manuscript that fully addresses all points raised by reviewer #1 (see below).   In particular, your explanation of the anisotropic growth mechanism in 3D that you added on lines 11-12 and 399-402 didn't seem to address the open question how the growing sheets (in 3D) remain (mostly) straight in the flow direction and only branch perpendicular. Your sheer stress hypothesis must somehow include a vector that cells are sensing and not just the magnitude. What biological processes could underlie such vectorial sensing? Reviewer #1 seems to draw a parallel to lung morphogenesis here.   Also, please check that your published source code is the current version and matches your manuscript's results and description. Currently, there are discrepancies as reviewer #1 points out. Please upload the used source code or correct your model description accordingly. 

We look forward to receiving your revised manuscript.

Kind regards,

Lutz Brusch, Ph.D.

Academic Editor

PLOS ONE

Journal Requirements:

Reviewers' comments:

Reviewer's Responses to Questions

**Comments to the Author**

1. If the authors have adequately addressed your comments raised in a previous round of review and you feel that this manuscript is now acceptable for publication, you may indicate that here to bypass the “Comments to the Author” section, enter your conflict of interest statement in the “Confidential to Editor” section, and submit your "Accept" recommendation.

Reviewer #1: (No Response)

Reviewer #2: All comments have been addressed

2. Is the manuscript technically sound, and do the data support the conclusions?

Reviewer #1: Partly

Reviewer #2: Yes

3. Has the statistical analysis been performed appropriately and rigorously? 

Reviewer #1: N/A

Reviewer #2: N/A

4. Have the authors made all data underlying the findings in their manuscript fully available?

Reviewer #1: Yes

Reviewer #2: Yes

5. Is the manuscript presented in an intelligible fashion and written in standard English?

Reviewer #1: Yes

Reviewer #2: Yes

6. Review Comments to the Author

Reviewer #1: The authors have done minimal changes necessary to address most of my minor points just enough, which has improved the quality of the manuscript, but they have left major concerns largely untouched.

There is still a lack of discussion of limitations of the 2D model and its links (whether absent or not) to anisotropic growth mechanisms in 3D. I think it is fine to make assumptions and propose a working principle while the actual biology is not fully understood, as the authors have done here. But this should then at least be discussed appropriately, especially since it is difficult to reconcile the 2D growth model with self-avoidance with a mechanistic way to grow bone tissue anisotropically in 3D. How do the authors picture the growth process in 3D? In 2D it is easy to place new particles at the tips of a tree, but how would the bone sheets grow analogously in 3D? Could the proposed mechanosensing of wall shear stress from fluid flow act as a cue in which direction to grow? I don’t expect the authors to have answers to these questions, but they need to be discussed at least, especially since similar ideas on growth anisotropy from wall shear stress sensing in the developing airway have appeared already in the literature (Conrad et al., Development 148, dev194209, 2021).

Moreover, now that the mathematical model description has been revised, it seems that the code implementation does not match the description in the manuscript. From Surface.h, it looks like Newtonian dynamics (a second-order differential equation) is solved with the semi-implicit Euler method. When params.damping = 0, then (essentially)

position += acceleration * dt * dt;

where acceleration is f in the manuscript. From the manuscript, however, one gets a different impression, namely that a discrete form of a first-order differential equation is implemented. Also, I am unable to spot the exponentially decaying term k(t) in the code. There isn’t a single call of the exponential function. My guess is that the authors just overdamped their model with supercritical damping coefficients (i.e., small dt in their implementation?), but that would NOT be the same as an exponentially decaying k(t) prefactor in the acceleration or force. k(t)~exp(t/tau) would model plasticity, whereas overdamping models viscosity – two qualitatively different concepts. The description in the manuscript needs to be revised substantially to match what has really been implemented, with a specification of the proper equations of motion and all model parameters used. This also links to my previous comment that the first manuscript version named the f terms “forces”, which they aren’t in the formulas presented. This has now been reworded, but only in a minimal sense that will make it even harder for the readers to follow, as the now-called “displacements” are still named f, while the revised manuscript still speaks of “balance between the relevant forces” and “spring forces”. This is all subpar and unnecessarily confusing.

Coming back to the tree area A_b. I had to look up the code to understand the author’s response to my and referee 2’s comment on the lack of clarity on how the tree length is converted to an area. I think it is still not clear in the manuscript. Apparently, the actual tree length is used as the occupied area. While this may work numerically, it makes no sense physically, as the units don’t match. There is a dimensional proportionality factor missing – even if it is one, in simulation units of length. This should still be specified in the manuscript (potentially in units of r_0, which sets the simulation length scale) to allow the readership to follow what was done.

Finally, I still perceive the claims about similarity of distributions in lines 272, 290 as overly strong. I understand that the claims were made based on the observed standard deviations and RMSE. But this is insufficient to claim similarity. Two distributions can be vastly different yet have the same standard deviations. Why not just comment on similarity of the standard deviations and/or means, rather than the entire distributions? Such a claim would then be justified.

Reviewer #2: All comments by the referees have been adequately addressed and the paper is now ready to be published.

7. PLOS authors have the option to publish the peer review history of their article (what does this mean?). If published, this will include your full peer review and any attached files.

Reviewer #1: No

Reviewer #2: No

---

## [Editor Report · Decision Letter 2]

16 Dec 2024

A model for maxilloturbinate morphogenesis in seals

PONE-D-24-24012R2

Dear Dr. Kings,

Thank you for having addressed and answered all follow-up comments by Reviewer 1. This has clarified the presentation of the methods and results.

We’re pleased to inform you that your manuscript has been judged scientifically suitable for publication and will be formally accepted for publication once it meets all outstanding technical requirements.

Kind regards,

Lutz Brusch, Ph.D.

Academic Editor

PLOS ONE

---

## [Editor Report · Acceptance letter]

PONE-D-24-24012R2

PLOS ONE

Dear Dr. Kings,

I'm pleased to inform you that your manuscript has been deemed suitable for publication in PLOS ONE. Congratulations! Your manuscript is now being handed over to our production team.

Kind regards,

on behalf of

Dr. Lutz Brusch

Academic Editor

PLOS ONE